# Imaging gigahertz zero-group-velocity Lamb waves

Qingnan Xie[1,4], Sylvain Mezil[2,4], Paul H. Otsuka[2], Motonobu Tomoda[2,5], Jérôme Laurent[3], Osamu Matsuda [2,5], Zhonghua Shen[1] & Oliver B. Wright [2,5]

Zero-group-velocity (ZGV) waves have the peculiarity of being stationary, and thus locally confining energy. Although they are particularly useful in evaluation applications, they have not yet been tracked in two dimensions. Here we image gigahertz zero-group-velocity Lamb waves in the time domain by means of an ultrafast optical technique, revealing their stationary nature and their acoustic energy localization. The acoustic field is imaged to micron resolution on a nanoscale bilayer consisting of a silicon-nitride plate coated with a titanium film. Temporal and spatiotemporal Fourier transforms combined with a technique involving the intensity modulation of the optical pump and probe beams gives access to arbitrary acoustic frequencies, allowing ZGV modes to be isolated. The dispersion curves of the bilayer system are extracted together with the quality factor $Q$ and lifetime of the first ZGV mode. Applications include the testing of bonded nanostructures.

[1] School of Science, Nanjing University of Science and Technology, 210094 Nanjing, People's Republic of China. [2] Division of Applied Physics, Graduate School of Engineering, Hokkaido University, Sapporo 060-8628, Japan. [3] Institut Langevin, ESPCI ParisTech, CNRS, 1 rue Jussieu, 75238 Paris, Cedex 05, France. [4]These authors contributed equally: Qingnan Xie, Sylvain Mezil. [5]These authors jointly supervised this work: Motonobu Tomoda, Osamu Matsuda, Oliver B. Wright. Correspondence and requests for materials should be addressed to S.M. (email: sylvain.mezil@eng.hokudai.ac.jp)

Waveguides channel propagating waves with reduced losses thanks to the confined dimensions, and are widely used in optics and acoustics. They are dispersive, i.e., the phase and the group velocities differ. Zero-group-velocity (ZGV) modes are particular points in a dispersion relation where the group velocity vanishes whereas the phase velocity remains finite. These modes can be found in most waveguide geometries (e.g., fibres and cylinders[1,2], plates[3,4], etc.) and have the advantages of combining reduced losses and high Q factor. Many applications take advantage of these unique properties. In optics, they are, for example, implicated in soliton propagation, pulse compression and microcavity confinement[1,5,6]. In acoustics, they are implicated in structural testing using Lamb waves, i.e., in geometries with free-surface boundary conditions and for which the acoustic wavelength is of the same order as the thickness[7]. These ZGV Lamb modes offer, for instance, methods for estimating the Poisson's ratio[8], thin layer thicknesses[9], elastic constants[10,11], and interfacial stiffnesses between bonded plates[12,13], or fatigue damage[14]. ZGV Lamb modes have their energy trapped within a specific lateral region, offering a local measurement. These modes can be accessed by contactless excitation and detection, achievable with air-coupled transducers[3], electromagnetic acoustic transducers[15] or lasers[4,8–14], often working in the kHz–MHz range.

The observation of ZGV Lamb modes in plates has been extended up to the GHz range by the use of interdigital transducers[16] or intensity-modulated continuous lasers[17], but only up to ~40 MHz by the use of pulsed lasers[14]. Observations of GHz ZGV Lamb waves with pulsed lasers has, however, not proved possible owing to the extremely sharp resonances associated with ZGV modes—exhibiting Q factors up to 14,700[18], for example—whereas acoustic frequencies are usually limited to integral multiples of the laser repetition rate. Ultrashort-pulse lasers are ideal for time-domain imaging, but to our knowledge the imaging of ZGV modes in two dimensions has not been investigated.

In this paper, we image a GHz ZGV Lamb mode in a bilayer consisting of a silicon-nitride plate coated with polycrystalline titanium by means of a time-resolved two-dimensional (2D) imaging technique incorporating an ultrashort-pulse laser[19]. We overcome the above-mentioned frequency limitation by the use of arbitrary-frequency control that takes advantage of the sidebands introduced by additional intensity modulation in the laser beam paths[20,21]. In particular, we identify and isolate the first ZGV mode, its associated Q factor and its lifetime. The experimental dispersion curves of the bilayer are obtained, clearly showing the location of the ZGV mode in frequency-wavevector space and its acoustic energy localization.

## Results

**Experimental setup and theoretical model**. The sample, depicted in Fig. 1a, consists of a square silicon-nitride plate (of approximate composition $Si_3N_4$) of thickness 1830 nm coated with a 660 nm sputtered polycrystalline titanium film (see Supplementary Note 2). Experiments were carried out with an optical pump-and-probe technique combined with a common-path Sagnac interferometer, that allows the possibility of imaging[22]. A pump optical beam focused to a micron-sized region of the sample surface $(x, y)$ is used to excite Lamb waves, and the temporal or spatiotemporal evolution of the normal surface particle velocity is obtained with ~100 ps time-resolution with a micron-sized probe-beam spot and by varying the pump-probe delay time with a delay line. Deformations occur throughout the sample thickness (as expected for Lamb waves). For the initial experiments, the pump and probe beams are co-focused to one point on the sample, and for the later experiments the probe beam is scanned over the surface in 1D or 2D. The arbitrary-frequency technique involving the modulation of the pump beam and/or probe beam is described in detail in Methods.

In order to facilitate the identification of a ZGV mode, we calculate the acoustic dispersion relation of our bilayer structure, making use of literature elastic constants[23] as well as layer thicknesses obtained from ultrafast pulse-echo measurement (see Supplementary Notes 1 and 2 for details). The first ZGV Lamb mode is predicted to be at frequency $f_1^{th} = 1.7248$ GHz and wavenumber $k_1^{th} = 0.620\ \mu m^{-1}$. Two other ZGV Lamb modes below 10 GHz are predicted near 3 and 7 GHz (see Supplementary Table 1). We concentrate in this paper on the first ZGV Lamb mode, which has a significant out-of-plane acoustic displacement component (see Supplementary Fig. 2).

**Detection of a GHz ZGV Lamb mode**. Figure 1b shows the temporal evolution of the out-of-plane surface particle velocity at an acoustic frequency of $f = 1.6900$ GHz for co-focused pump and probe spots of a few microns in width on the sample (see Methods). This frequency was isolated initially by the help of the above theoretical prediction for the first ZGV resonance and subsequently by frequency tuning. A thermal background variation is subtracted using a second-order polynomial function, obtained by the least-squares method (as with all subsequent fits).

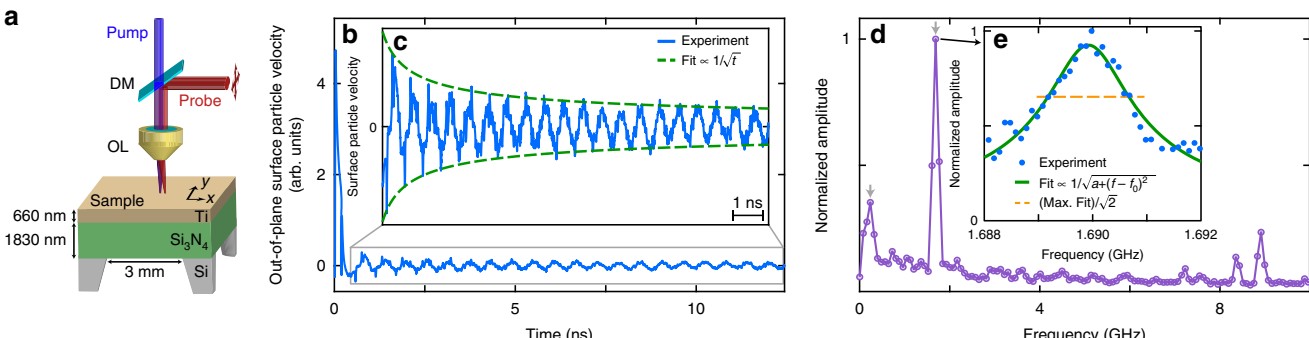

**Fig. 1** Sample, surface particle velocity variation, and acoustic frequency spectrum. **a** Schematic diagram of the sample. DM: dichroic mirror, OL: objective lens. The double-ended arrows represent the probe-beam 2D scanning realized by moving mirrors and a confocal lens pair (see Methods). **b** Out-of-plane surface particle velocity measured for co-focused pump and probe beams. **c** Zoom-in of the ZGV Lamb mode amplitude temporal evolution and fit $\propto 1/\sqrt{t}$. **d** Frequency spectrum obtained after temporal Fourier transform. Arrows indicate the two modes imaged in Fig. 2. **e** Evolution of the normalized amplitude in the frequency window $1.688 \leq f \leq 1.692$ GHz. The solid line is a fit to the experimental data in the form of the square root of a Lorentzian. The dashed line represents $1/\sqrt{2}$ (i.e., −3 dB) of the maximum amplitude of the fit

This clearly reveals the ZGV resonance, a mode with a lifetime greater than the laser repetition period of ~12 ns (as explained in detail later). An enlarged view of the data is shown in Fig. 1c. Two effects contribute to the decay in amplitude. The first originates from the second-order term in the dispersion relation $\omega(k)$ in the vicinity of the ZGV resonance, where $\omega$ is the angular frequency and $k$ the acoustic wavenumber. This produces a $1/\sqrt{t}$ decrease with time $t$, as previously observed at lower frequencies[24]. The second originates from viscoelastic attenuation, yielding a decay $\propto e^{-t/\tau_0}$, where $\tau_0$ is the mode lifetime, and can be observed only after a certain time has passed. The amplitude decrease in the present case can be fitted to a good approximation with a $1/\sqrt{t}$ function, as shown in Fig. 1c. For our time window of ~20 periods for the ZGV mode, no exponential decrease is visible, precluding a derivation of $\tau_0$ by this method.

The experimental spectrum obtained from the combination of the different scanned frequencies is displayed in Fig. 1d (see Methods for details). Besides the strong ZGV resonance at 1.6900 GHz, other peaks are observed at 0.2391, 7.2365, 8.3594, and 8.9202 GHz, which are selected (a) owing to the choice of pump modulation frequencies, as shown in Methods, (b) because with co-focused optical pump and probe spots one only detects modes that do not leave the excitation region, i.e. modes with zero or low group velocity, and (c) because with the probe-laser normal incidence only modes with significant normal displacement are detected. The detected modes lie on the $qA_0$, $qA_4$, $qS_8$ and $qS_5$ branches of the acoustic dispersion relation, where $qA$ refers to quasi-antisymmetric and $qS$ refers to quasi-symmetric Lamb waves (see Supplementary Note 2). The observed ZGV mode thus corresponds to the $qS_1$ ZGV.

The experimental ($f_1^{\text{exp}} = 1.6900\,\text{GHz}$) and theoretical ($f_1^{\text{th}} = 1.7248\,\text{GHz}$) ZGV frequencies are in reasonable agreement. The residual ~2 % mismatch may be caused either by a difference in the $Si_3N_4$ layer parameters (see Supplementary Note 2) or to imperfect adhesion between the two layers[12,13]. The two other ZGV Lamb modes, predicted near 3 and 7 GHz, could not be detected. This is thought to be owing to a mismatch between the probed frequencies and the ZGV frequencies (see Methods) as well as, for the 7 GHz peak, to the smaller out-of-plane surface displacement expected for this mode (see Supplementary Fig. 2).

Figure 1e shows the amplitude–frequency relation around $f_1^{\text{exp}}$. In order to extract the associated quality factor $Q$, we fit with the square root of a Lorentzian function, $b/\sqrt{a + (f - f_0)^2}$, where $a$, $b$ and $f_0$ are fitting parameters, yielding $f_0 = 1.68992\,\text{GHz}$ and $Q = 1150$. This of the same order of magnitude as the Q factors previously reported for thin composite silicon-nitride/metal/oxide plates suspended by thin beams and driven in thickness longitudinal resonances at similar frequencies[25]. For ZGV modes, Q factors up to 14,700 at MHz frequencies have been observed in other materials[18,26], values which depend strongly on the ultrasonic attenuation at the frequency in question, as discussed in the next section, as well as parameters such as the possible imperfect adhesion between the two layers.

**Imaging a ZGV Lamb mode.** The spatiotemporal evolution of the acoustic field is obtained by scanning the probe spot in 2D over a $55 \times 55\,\mu\text{m}^2$ area using 301 frames (i.e., images obtained at different pump-probe delay times—see Methods). We specifically target the ZGV mode (at $f_1^{\text{exp}}$) and the branch $qA_0$ at 0.24 GHz, indicated by the two downward-pointing arrows in Fig. 1d. To eliminate unwanted frequency components, filtering of small and high wavenumbers is conducted in Fourier space. We thereby extract the amplitude field, as shown in Fig. 2a, b for $f = 0.2391$ and $f_1^{\text{exp}} = 1.6900\,\text{GHz}$, respectively (also viewable as animations in the Supplementary Movie 1). The 1.6900 GHz data represents, to our knowledge, the first experimental 2D movie of a ZGV mode, extending the 1D observations of Laurent et al.[27]. Comparing the animations, one can immediately ascertain that the ZGV mode is not propagating, in striking contrast to the $qA_0$ branch, which corresponds to propagating modes.

For a single-frequency point-excited wave in two dimensions, the radial form of the out-of-plane displacement is proportional to $J_0(kr)$, where $J_0$ is the first-order Bessel function and $r$ is the radial distance (see, e.g., Wright et al.[28]). This has previously been confirmed by theory and by simulation for ZGV Lamb modes[17,24]. Figure 2c shows a fit to our data at 1.6900 GHz with the function $J_0(kr)$, which gives reasonable agreement, also clear from the cross section shown in Fig. 2d. The value of $k$ for the best fit, $k_1^{\text{exp}} = 0.55 \pm 0.13\,\mu\text{m}^{-1}$, is in fair correspondence with the predicted value $k_1^{\text{th}} = 0.620\,\mu\text{m}^{-1}$ from the theoretical dispersion relation. (The fit does not take into account the smoothing due to the finite spot diameters, but we verified that its effect has a negligible influence on the fitted curve.) One can notice that the experimental amplitude on the first main side lobes is slightly larger than the theoretical Bessel fitted function, which is in agreement with the results obtained by Laurent et al.[27]. Away from the central peak, the experimental amplitude is lower; these effects can be attributed to high-frequency material losses.

Our measurement of the Q factor ($Q = 1150$) and wavenumber ($k_1^{\text{exp}} = 0.55\,\mu\text{m}^{-1}$) for the ZGV mode at 1.6900 GHz allows an estimate of the lifetime $\tau_0$. As a first step, for a free-standing layer of a single isotropic material and considering only viscoelastic

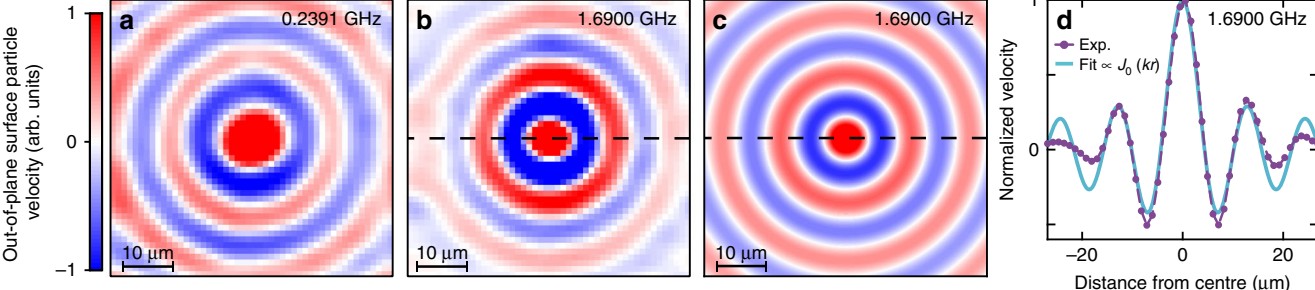

**Fig. 2** Imaging a ZGV Lamb mode. **a**, **b** Normalized images of the measured out-of-plane surface particle velocity at **a** 0.2391 and **b** 1.6900 GHz. The x and y axis directions are shown in Fig. 1a. Animations are viewable in the Supplementary Movie 1. **c** 2D acoustic field at 1.6900 GHz based on a fit to a Bessel function $J_0(kr)$. **d** Normalized out-of-plane surface particle velocity evolution of the experimental (dots) and fitted (solid line) data as a function of radial distance for a frequency of 1.6900 GHz (dashed lines in **b** and **c**)

losses[18,24], the spatial attenuation coefficient is given by

$$\alpha = \frac{k}{2Q}, \qquad (1)$$

where $\alpha$ is an effective attenuation coefficient (in $m^{-1}$). This relation yields $\alpha = 240\ m^{-1}$ for the above-mentioned ZGV mode. One can then make use of the relation $\tau_0 = 1/(\alpha v_p)$, where $v_p$ is the phase velocity, to find $\tau_0 \approx 0.22\ \mu s$. This is much longer than the time ~10 ns for an acoustic mode with a typical sound velocity ~5 km.s$^{-1}$ to leave the imaged region of $55 \times 55\ \mu m^2$ in which the ZGV amplitude is significant.

The value of $\alpha$ obtained can be compared with other high-frequency measurements. Assuming an $f^2$ variation in $\alpha$, as expected from viscous losses, our value $\alpha = 240\ m^{-1}$ lies between those extrapolated for longitudinal waves in silicon nitride (29 m$^{-1}$) and polycrystalline titanium (2300 m$^{-1}$) to a frequency of 1.69 GHz[29,30]. A detailed comparison is difficult because ZGV modes couple both longitudinal and transverse strain components, which are distributed over the two layers (see Supplementary Fig. 2).

**Dispersion curves**. More complete information is available with an experimental knowledge of the Lamb-wave dispersion curves. To this end, the pump beam is focused to a micron-sized line source with a cylindrical lens (see Methods). The probe-beam spot is scanned along the direction $x$ (>0, see Fig. 1a) perpendicular to this line source over distances up to 100 μm in 0.05 μm steps to obtain the spatiotemporal variation of the acoustic field. A 2D Fourier Transform (FT) (temporal FT and 1D spatial FT) allows the extraction of the acoustic dispersion curves, as shown in Fig. 3a for both experiment and theory, which show good agreement for the modes visible in experiment (bright regions). The positive wavenumbers in Fig. 1 correspond to waves with phase velocity along the $+x$ direction. The three branches $qA_0$, $qS_0$, and $qA_1$, observed here below ~2 GHz, are detected for a wide range of wavenumbers, unlike branches with higher frequencies, which are detected only for small wavenumbers $k$. The first ZGV mode is clearly observed at $k = 0.54 \pm 0.03\ \mu m^{-1}$, in agreement with the value extracted in the 2D-scan experiment. The second ZGV mode, predicted at $f_2^{th} = 3.0024\ GHz$ and $k_2^{th} = 0.732\ \mu m^{-1}$ is evident at $f_2^{exp} = 2.972\ GHz$ and $k_2^{exp} = 0.83 \pm 0.03\ \mu m^{-1}$, in fair agreement. This mode may not have been observed in the previous experiment with co-focused laser beams owing to the different excited wavevector spectrum $\left(k_1^{th} < k_2^{th}\right)$. Other peaks that were previously detected at 0.2391, 7.2365, and 8.9202 GHz are again evident on the branches $qA_0$, $qA_4$, and $qS_5$, respectively. (The mode previously detected at 8.3594 MHz is absent here, and no corresponding mode appears on the dispersion relation. Its previous appearance in the spectrum of Fig. 1d seems likely to be an experimental artifact.) Some branches, such as $qA_2$ near 3.6 GHz as well as $qS_0$ and $qA_1$, are revealed only in this new measurement. We attribute residual discrepancies between the theoretical and experimental dispersion curves to uncertainties in the elastic parameters and in the layer thicknesses, and to the possible imperfect layer bonding[31].

The dispersion curves in Fig. 3b are a magnified view of Fig. 3a, this time including negative wavenumbers corresponding to $-x$-directed propagation. The main features are as follows:

1. for the $qA_0$, $qS_0$ and $qA_1$ branches and the region $(\omega, k)$ with $|k|$ larger than that for the $qS_1$ ZGV (the latter indicated by a downward-pointing arrow in Fig. 3a), the intensity is much greater for $k > 0$;

2. for the region $(\omega, k)$ with $|k|$ smaller than that for the $qS_1$ ZGV, the intensity is much greater for $k < 0$;

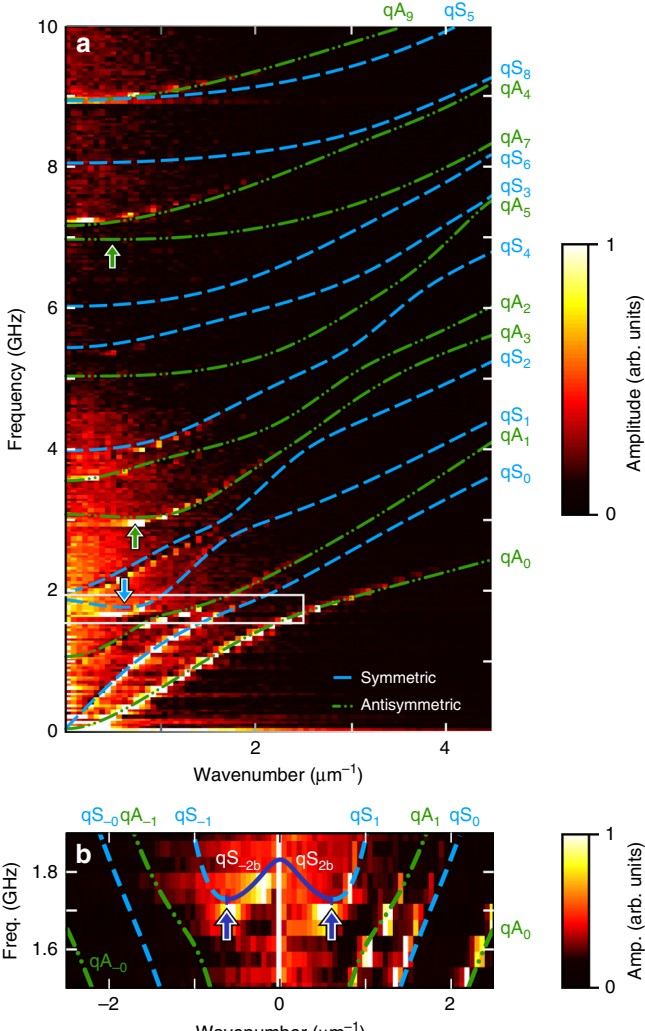

**Fig. 3** Dispersion curves of the bilayer. **a**, **b** Experimental (image) and theoretical (blue dashed and green dotted-dashed lines) dispersion curves for **a** $0 \le f \le 10$ GHz and $0 \le k \le 4.5\ \mu m^{-1}$ **b** at frequencies near the first ZGV mode for $1.5 \le f \le 1.9$ GHz and $-2.5 \le k \le 2.5\ \mu m^{-1}$. The solid navy blue section of the dispersion curves corresponds to the portions where the phase and group velocities are anti-parallel. **a** The box corresponds to the enlarged region in **b** for $k > 0$. **a**, **b** Arrows indicate ZGV points. Plots are independently normalized

3. for the region $(\omega, k)$ in the vicinity of the $qS_1$ ZGV, near wavenumber $k_1$ and frequency $f_1$, the intensities for $k > 0$ and $k < 0$ are comparable.

## Discussion

Because of the symmetry of the excitation with respect to coordinate $x$, acoustic waves with positive and negative $k$ should be generated with equal amplitude and initially form a single wave packet located at the excitation point. This wave packet is eventually broadened and split because of the acoustic dispersion and the broad distribution of positive and negative values of $k$. Its wave components are plane-wave modes spreading over all 2D space, but in general, their sum is observable as a wave packet only if their phase constructively interferes. In other words, there are two possibilities for observing no acoustic field: a complete lack of such modes or the overlapping of several modes with random phase. (Compare the Fourier transform of a δ-function:

it contains an infinite number of fully spreading plane waves, but is only finite at a point in space.) Since we observe the waves in the region $x > 0$, the waves we can primarily observe are restricted to those with $v_g \geq 0$. Waves with $v_g < 0$ exist, but give negligible contribution to the waves we observe because we only probe for positive $x$ values over a region with little overlap with the pump spot.

The feature (1) above can therefore be attributed to the waves having parallel group velocity $v_g$ and phase velocity $v_p$. With reference to the dispersion curves, all $k$ values involved in feature (1) show positive group velocity $v_g > 0$. On the other hand, feature (2) can be attributed to the waves having anti-parallel $v_g$ and $v_p$; the $k$ values involved in feature (2) can be seen to be negative, but are associated with a positive slope on the $\omega-k$ plot, i.e., they possess $v_g > 0$ but with $v_p < 0$. The feature (3) can be attributed to the waves with zero or very small group velocity. The wave packet consisting of these near-zero or zero-$v_g$ wave components remains almost stationary, and both positive and negative $k$ components contribute to this wave packet over an extended period of time. These features are consistent with those of Philippe et al.[32]. In Fig. 3b, we label the part of the $+k$ $qS_1$ branch with the negative slope as $qS_{2b}$ (and $qS_{-2b}$ for the equivalent $-k$ part), in agreement with its regular appellation[33,34], where 'b' stands for backward wave. All other branches with finite positive group velocity are observed only for $k > 0$, with the exception of $qS_5$ near 9 GHz, for which $v_g$ is small for low $k$, and this branch is observed for both positive and negative $k$ with similar amplitude. Only the branch containing the ZGV point at 1.6900 GHz has a greater amplitude for negative $k$ compared with that for positive $k$.

In conclusion, we image a zero-group-velocity Lamb mode in the time domain. We apply an ultrafast time-domain technique with arbitrary GHz-acoustic frequency control to a nanoscale bilayer consisting of a silicon-nitride plate coated with a titanium film to provide this observation at unprecedented frequencies in the GHz range. Our combination of both time-domain 1D and 2D optical scanning methods provides a comprehensive probe for the ZGV dynamics. We isolate the $qS_1$ ZGV mode at ~1.7 GHz and probe its spatial and temporal characteristics, including its $Q$ factor ~1000. The spatial form of this quasi-point-excited ZGV is directly verified by Fourier analysis to correspond closely to the expected Bessel function in 2D, allowing its wavenumber to be derived and its ZGV nature to be directly verified in the time domain from its relatively long lifetime ~0.2 μs. Experimental dispersion curves of this bilayer system are also obtained, and show good agreement with a theoretical model and reveal another ZGV mode.

Applications of real-time imaging of ZGV Lamb modes include the detection of defects in adhesion and deviations in interfacial stiffness, so this high-frequency imaging technique should provide new avenues for evaluating and quantifying the mechanical integrity of nanostructures. In particular, our approach allows the probing of nanolayers. One example is the measurement of interfacial adhesion in two- or multi-layer nanomembrane systems. ZGV modes have already proven efficient in the determination of interfacial adhesion in layers of thickness in the mm range[12,13], so our work should encourage research into the evaluation of both normal and tangential interfacial adhesion in ultrathin multilayers of thickness in the nm range, complementing other GHz approaches to nanolayer adhesion metrology based on picosecond ultrasonics[35–37].

Moreover, since the use of finely-controlled arbitrary frequencies lends itself so well to the precise determination of resonances and their associated $Q$ factors, it should prove possible to apply GHz ZGV imaging to the characterization of elastic constants in nanoscale layers in a similar way to that demonstrated in picosecond ultrasonics[38]. Our 2D imaging approach

would also be particularly powerful for analysing complex directional behaviour when applied to nanoscale-thickness anisotropic layers—following the example of previous time-domain GHz surface-acoustic-wave imaging of crystals and phononic crystals[39,40]—including the possibility of directional excitation with laser line sources. The ability to image and thereby directionally distinguish and analyse the different in-plane ZGV acoustic modes would further facilitate the extraction of elastic constants on the nanoscale by analogy with work on anisotropic ZGV modes in the MHz frequency range[10].

Comparing our approach with that of surface-acoustic-wave transient gratings in the same GHz frequency range (see, for example, Bruchhausen et al.[41]) on the same sample would be fruitful, but especially our technique, exhibiting a much finer frequency control, should allow higher resolution of individual resonances and the detection of modes with larger $Q$ factors. Finally, by extending the 2D time-domain imaging methods used for light propagation in optical waveguides and microstructures[42], it should be possible to image photonic ZGV modes in the time domain in a way analogous to the work presented here.

## Methods

**Measurement setup.** A Ti:sapphire pulsed laser produces a series of ~100 fs width pulses at repetition rate $f_{rep}$ = 80.38 MHz and wavelength 830 nm. Part of this beam is frequency doubled ($\lambda$ = 415 nm) and used for the pump with a pulse energy of ~0.15 nJ. It is intensity modulated with an acousto-optic modulator (AOM) at frequency $f_p$. For experiments with a circular pump spot, the pump beam is focused on the Ti film side of the sample at normal incidence with a ×50 objective lens to a 4.2 μm radius (at $1/e^2$ intensity) to optimize the first ZGV Lamb mode generation (see Supplementary Note 2). This generates, in the centre of the pump spot, an instantaneous temperature rise of ~80 K after each laser pulse and a steady state temperature rise of ~50 K (see Supplementary Note 3). The Lamb waves thermoelastically generated with our pump are calculated to have an amplitude of ~10 pm. To determine the dispersion relation experiments are also carried out using a pump spot in the form of a line of $1/e^2$ intensity half-width 1.5 μm and a length of 5 μm.

The other part of the beam, at $\lambda$ = 830 nm, is used for the probe. It is focused to a 2.8 μm radius (at $1/e^2$ intensity) for both types of pump beam focusing with the same objective lens as used for the pump and with a ~0.03 nJ pulse energy. When required, it is intensity modulated with another AOM at the frequency $f_s$. The modulation frequency is set to allow heterodyne detection within the 3 MHz photodetector bandwidth $\left(|f_p - f_s| \leq 3\,\text{MHz}\right)$. A Sagnac interferometer is incorporated[22], producing two probe pulses separated by a time interval of $t < 200$ ps. A 2D spatial scanning system for the probe beam, composed of two moving mirrors in a confocal lens pair configuration[22], is mounted in the path of the probe beam together with a delay line to monitor the spatiotemporal evolution of the acoustic field up to ~10 GHz. However, modes are only detected with wavelengths $\lambda$ sufficiently large to be resolved by the finite size of the probe spot, i.e., $\lambda \gtrsim 2$ μm. (The effect of the pump spot on the generated acoustic wavelength spectrum is described in detail in the Supplementary Note 2). The out-of-plane surface particle velocity modulates the optical phase, which is converted to an intensity modulation that is detected with a lock-in amplifier. The arbitrary-frequency technique allows one to access frequencies $nf_{rep} + mf_p$, $n$ an integer and $m = \pm 1$, by recording both in-phase and quadrature components[20,21,43].

**Data analysis.** The setup used here makes use of the following: double modulation (i.e., intensity modulation of both pump and probe beams), a delay set in the probe-beam line, and a probe modulation upstream of the delay. The arrangement is described in detail in Matsuda et al.[21]. Because of the pump intensity modulation, the excited frequencies are $nf_{rep} + mf_p$, where $m = +1$ for the upper sideband and $-1$ for the lower sideband. The generated acoustic field can be expressed as a superposition of these frequency components as

$$S(t) = \sum_{n,m} A_{n,m} \cos[-(n\omega_{rep} + m\omega_p)t + \phi_{n,m}], \quad (2)$$

where $A_{n,m}$ and $\phi_{n,m}$ are the position dependent amplitude and phase of a vibrational mode ($n$, $m$) of frequency $nf_{rep} + mf_p$. In the summation of Eq. (2), the case ($n$, $m$) = (0, −1) is not included. From the detection system (i.e., a photodetector connected to a lock-in amplifier), we access the in-phase ($X$) and quadrature ($Y$) components of the lock-in output, which are proportional to the acoustic out-of-plane surface particle velocity. (We assume $X \propto \int u(t) \cos\omega_{ref}\, t\, dt$ and $Y \propto \int u(t) \sin\omega_{ref}\, t\, dt$ where $u(t)$ is the signal from the photodetector and $\omega_{ref} = |\omega_s - \omega_p|$.) Both are functions of delay time $\tau$ and probe spot position.

The complex signal $Z \equiv X + iY\mathrm{sgn}(\omega_p - \omega_s)$ can be rewritten as

$$Z(\tau) = \sum_{n,m} A_{n,m} e^{im[-(n\omega_{rep} + m\omega_s)\tau + \phi_{n,m}]}. \tag{3}$$

In the summation of Eq. (3), again the case $(n, m) = (0, -1)$ is not included. The amplitude and the phase are found from a Fourier transform (FT):

$$A_{n,m} e^{im\phi_{n,m}} = \frac{1}{T} \int_0^T Z(\tau) e^{i(mn\omega_{rep} + \omega_s)\tau} d\tau. \tag{4}$$

It is then possible to analyse the data at an angular frequency $\omega$ as a real-valued term involved in Eq. (2), i.e., giving a spatial pattern of the vibration at $\omega = 2\pi(nf_{rep} + mf_p)$. Modifying the frequency $f_p$ thus allows arbitrary acoustic frequency control.

In the first experiment, the pump frequency $f_p$ is increased from 0.1 up to 4 MHz in steps of 0.1 MHz. For each pump frequency we perform a single delay line scan for an acquisition time ~2 min. This provides data at frequencies $nf_{rep} + mf_p$. In the spectrum displayed in Fig. 1e, one can see the amplitude evolution associated with $n = 21$ and $m = +1$ for all these pump frequencies. The maximum amplitude is obtained for $f_p = 2.0$ MHz. In the following experiments the pump frequency is maintained at $f_p = 2.0$ MHz to optimize the ZGV Lamb mode generation. The resulting measured acoustic frequencies are $nf_{rep} \pm 2.0$ MHz. In Fig. 1d, the analysis covers all the scanned frequencies, but for each $n$ only the maximum amplitude is displayed for clarity (e.g., for $n = 21$ only the point associated with $f_p = 2.0$ MHz is displayed).

In the second experiment related to ZGV Lamb mode imaging, each frame of the animation has a spatial resolution of ~2 μm and is recorded in somewhat over 2 min. The data acquisition for a full animation requires ~12 h. The high number of frames (301) allows us to image at a rate of more than 10 frames per acoustic period modes with a frequency up to ~2.5 GHz.

In the final experiment to determine dispersion curves, each 1D spatial scan (of the probe beam for a given pump-probe delay) is performed in ~2 min, leading to a total acquisition time ~8 h and a spatial resolution ~2 μm. This time is considerably longer than in a typical surface-acoustic-wave transient grating experiment[44] owing to the need for the spatial scan in our case.

## Data availability
The data that support the findings of this study are available from the corresponding author upon reasonable request.

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

## Acknowledgements

We are grateful to Claire Prada and Alex Maznev for fruitful discussions. S. Mezil carried out this work as an International Research Fellow of the Japanese Society for the Promotion of Science (JSPS). We also acknowledge Grants-in-Aid for Scientific Research from the Ministry of Education, Culture, Sports, Science, and Technology (MEXT).

## Author contributions

O.B.W. and S.M. proposed the research goals and supervised the project. Q.X. performed experiments with the help of S.M. and M.T. The theoretical model and numerical model was developed by S.M. and J.L. Data were analysed by Q.X., S.M. and P.H.O. Theoretical support was provided by S.M., O.M., Z.S., and O.B.W. All authors helped prepare, critically review, and revise the manuscript.

## Additional information

**Competing interests:** The authors declare no competing interests.

