## [Peer Review File · Nature Communications]

Reviewers' comments:

Reviewer #1 (Remarks to the Author):

The authors use an ultrafast laser system to image Lamb waves propagating in a thin bilayer plate. The laser system permits arbitrary frequency control by taking advantage of the sidebands introduced by intensity modulation in the laser beam paths. The study focusses on the investigation of zero group velocity (ZGV) modes- points in a dispersion relation where the group velocity vanishes and the phase velocity remains finite. The two contributions of this manuscript are that (a) ZGV modes are investigated in a frequency range that is higher than previously reported using pulsed lasers and (b) time resolved 2D images of the displacement field are obtained. The imaging approach itself has been previously reported by the authors, but this is the first time that it has been used to examine ZGV modes.

The data presented in the manuscript and the conclusions reached by the authors are technically sound and convincing. In addition, the main contributions of the paper described above are novel, and the manuscript adds to the important body of recent work on ZGV modes in optics and acoustics. I believe that the manuscript will be of interest to those in the field. However, the results obtained for the high frequency ZGV modes follow the expected behavior previously observed by other authors at lower frequencies. The authors claim that the presented results on GHz ZGV modes may prove significant in evaluating and quantifying the mechanical integrity of nanostructures. I tend to agree with this statement but also believe that the authors need to do a better job in describing the significance of the results, and how the results will influence thinking in the field.

Reviewer #2 (Remarks to the Author):

The authors present the full experimental characterization of high-frequency (GHz) Lamb modes in substrates with zero group velocity. They use an all-optical pump and probe set up, in which the sample is excited by a pulsed laser and detected by the probe beam. It is a really challenging measurement, and they provided not only the detection of the mode, but also its spatio-temporal characterization. The paper is clearly written in general, and the results and discussion are properly justified.

I recommend this paper for publication in Nature Communications, but I'd ask the authors to clarify first some points:

1.- In page 2, when the authors explain the "Detection of a GHz ZGV Lamb mode" they say "A thermal background variation is subtracted using a polynomial function". ¿Could the authors explain this better or, at least, give a reference?"

2-Also in the same section, is Figure 1b the direct signal detection or average over some measurements? How many?

3.- In the "Imaging a ZGV Lamb mode" section, they say that they make a scan of 301 frames, what does it mean?

4. Also in the scan of the area, how long it takes to fully scan the area? What is the resolution?

5. For the obtention of the dispersion curves, they perform a 2D FT analysis by making sweep along the x axis. Is this faster than the transient grating method? Once more, how long it takes to make the measurement?

6. I would acknowledge in general more details about the acquisition times in the experiments.

7. For the spatial scan of the fields they have to move the probe, but in the set up they show in Fig.1a it is not clear to me how they can move the probe without moving the pump as well, which

obviously is not enough to do this characterization, could the authors be more clear in this point?

Reviewer #3 (Remarks to the Author):

Referee report

Manuscript ID: NCOMMS-18-36693

Title: "Imaging Gigahertz Zero-Group-Velocity Lamb Waves"

The authors report on the observation of zero-group-velocity Lamb modes in the hypersonic frequency range both in the time and space domain, by means of a state-of-the art ultrafast pump-probe optical microscopy. They exploit a smart pump and probe configuration allowing superior frequency resolution. The displacement field on a Ti-Si₃N₄ bilayer is spatially resolved with micrometric resolution. The zero-group-velocity Lamb mode life-time, Q-factor and dispersion relation are directly accessed.

The topic is timely and of high relevance. The results constitute a significant step forward in photoacoustic and bear great interests also for other communities. Indeed, I expect the methodology and the physics here shown will have a long-standing impact in fields such as mechanical nanometrology, non-destructive testing at the nanoscale and nanoscale-engineering of mechanical properties.

The manuscript is well written. The experimental evidence is exhaustive and its interpretation convincing. The data are clearly presented and thoroughly discussed. The authors furnish, through the SI and a tailored selection of the bibliography, all the elements needed to reproduce their work.

The manuscript fully complies with the criteria established for publication on Nature Communication. I therefore recommend publication of the present manuscript with minor revisions without need for further review.

Comments/suggestions.

I found the manuscript fully exhaustive both under a scientific and technical stand point. That said I have a couple of suggestions that might enlarge the interested audience.

In order to convey the significance of their message to other communities I suggest the authors to expand on their claim regarding their work potentials in the frame of mechanical metrology and non-destructive testing of nanomaterials, a claim which I strongly support.

The authors could:

(1) be more specific in addressing the "potential applications of real-time imaging of Gigahertz Zero-Group-Velocity Lamb Waves", going beyond the few lines reported at the end of page 6 - beginning of page 7.

(2) Surface Acoustic Waves (in their broad sense) in the tens of GHz range and beyond have been recently proposed as a formidable tool for ultrathin film nanometrology. It would be nice if the authors could comment on the possibility of using their technique in conjunction with SAWs as a tool for non-destructive testing of nanometric films or multilayers.

The x-y axis in Fig. 1 (a), which is referred to in the manuscript, is missing.

Reviewer #1 (Remarks to the Author):

The authors use an ultrafast laser system to image Lamb waves propagating in a thin bi-layer plate. The laser system permits arbitrary frequency control by taking advantage of the sidebands introduced by intensity modulation in the laser beam paths. The study focusses on the investigation of zero group velocity (ZGV) modes- points in a dispersion relation where the group velocity vanishes and the phase velocity remains finite. The two contributions of this manuscript are that (a) ZGV modes are investigated in a frequency range that is higher than previously reported using pulsed lasers and (b) time resolved 2D images of the displacement field are obtained. The imaging approach itself has been previously reported by the authors, but this is the first time that it has been used to examine ZGV modes.

The data presented in the manuscript and the conclusions reached by the authors are technically sound and convincing. In addition, the main contributions of the paper described above are novel, and the manuscript adds to the important body of recent work on ZGV modes in optics and acoustics. I believe that the manuscript will be of interest to those in the field. However, the results obtained for the high frequency ZGV modes follow the expected behavior previously observed by other authors at lower frequencies.

The authors claim that the presented results on GHz ZGV modes may prove significant in evaluating and quantifying the mechanical integrity of nanostructures. I tend to agree with this statement but also believe that the authors need to do a better job in describing the significance of the results, and how the results will influence thinking in the field.

Reply>> *We thank the reviewer for their comments. Referee 3 also made a similar suggestion for expanding the discussion of the significance. We have now greatly extended the conclusions, including new references, to better bring out the import of the results. The new conclusions are now as follows:*

*“Applications of real-time imaging of ZGV Lamb modes include the detection of defects in adhesion and deviations in interfacial stiffnesses, so this high frequency imaging technique should provide new avenues for evaluating and quantifying the mechanical integrity of nanostructures. In particular, our approach allows the probing of nanolayers. One example is the measurement of interfacial adhesion in two- or multi-layer nano-membrane systems. ZGV modes have already proven efficient in the determination of interfacial adhesion in layers of thickness in the mm range [Mezil et al., Appl. Phys. Lett. **105**, 021605, 2014; Mezil et al., J. Acoust. Soc. Am. **138**, 3202, 2015], so our work should encourage research into the evaluation of both normal and tangential interfacial adhesion in ultrathin multi-layers of thickness in the nm range, complementing other GHz approaches to nanolayer adhesion metrology based on picosecond ultrasonics [Rossignol et al., Phys. Rev. B **70**, 094102, 2003; Antonelli et al., MRS Bulletin **31**, 607, 2006; Grossman et al., New J. Phys. **19**, 053019, 2017].*

*Moreover, since the use of finely-controlled arbitrary frequencies lends itself so well to the precise determination of resonances and their associated Q factors, it should prove possible to apply GHz ZGV imaging to the characterization of elastic constants in nanoscale layers in a similar way to that demonstrated in picosecond ultrasonics [Mante et al., Appl. Phys. Lett. **93**, 071909, 2008]. Our 2D imaging approach would also be particularly powerful for analysing complex directional behaviour when applied to nanoscale-thickness anisotropic layers—following the example of previous time-domain GHz surface-acoustic-wave imaging of crystals and phononic crystals [Sugawara et al., Phys. Rev. Lett. **88**,*

185504, 2002; Otsuka et al., *J. Appl. Phys.* **117**, 245308, 2015]—including the possibility of directional excitation with laser line sources. The ability to image and thereby directionally distinguish and analyse the different in-plane ZGV acoustic modes would further facilitate the extraction of elastic constants on the nanoscale by analogy with work on anisotropic ZGV modes in the MHz frequency range [Cès et al., *J. Acoust. Soc. Am.* **132**, 180, 2012].

Comparing our approach with that of surface-acoustic-wave transient gratings (TGs) in the same GHz frequency range (see, for example, [Bruchhausen et al., *Phys. Rev. Lett.* **106**, 077401, 2011] on the same sample would be fruitful, but especially our technique, exhibiting a much finer frequency control, should allow higher resolution of individual resonances and the detection of modes with larger Q factors. Finally, by extending the 2D time-domain imaging methods used for light propagation in optical waveguides and microstructures [Engelen et al., *Nat. Phys.* **3**, 401 EP-, 2007], it should be possible to image photonic ZGV modes in the time domain in a way analogous to the work presented here.”

Reviewer #2 (Remarks to the Author):

The authors present the full experimental characterization of high-frequency (GHz) Lamb modes in substrates with zero group velocity. They use an all-optical pump and probe set up, in which the sample is excited by a pulsed laser and detected by the probe beam. It is a really challenging measurement, and they provided not only the detection of the mode, but also its spatio-temporal characterization. The paper is clearly written in general, and the results and discussion are properly justified.

I recommend this paper for publication in Nature Communications, but I'd ask the authors to clarify first some points:

1.- In page 2, when the authors explain the "Detection of a GHz ZGV Lamb mode" they say "A thermal background variation is subtracted using a polynomial function". Could the authors explain this better or, at least, give a reference? »

Reply>> *We thank the referee for the various comments. Answers are as follows: To make clear the background subtraction method we have added the following sentence. As this is standard in the field of transient reflectance measurements we think a reference is probably not required.*

“A thermal background variation is subtracted using a second-order polynomial function, obtained by the least-squares method.”

2.-Also in the same section, is Figure 1b the direct signal detection or average over some measurements? How many?

Reply>> *The signal presented in Fig 1b corresponds to direct signal detection. In the section Methods, Data analysis, we have modified the text as follows:*

“In the first experiment, the pump frequency f_p is increased from 0.1 up to 4 MHz in steps of 0.1 MHz. For each pump frequency we perform a single delay line scan for an acquisition time ~ 2 min.”

3.- In the "Imaging a ZGV Lamb mode" section, they say that they make a scan of 301 frames, what does it mean?

Reply>> *The number of frames correspond to the number of images acquired at different pump-probe delays (by scanning the delay line). Our sentence has been modified as follows to address this point:*

"The spatiotemporal evolution of the acoustic field is obtained by scanning the probe spot in 2D over a $55 \times 55 \mu\text{m}^2$ area using 301 frames (i.e., images obtained at different pump-probe delay times— see Methods)."

And in Methods section we also added the following:

"The high number of frames (301) allows us to image at a rate of more than 10 frames per acoustic period modes with a frequency up to ~ 2.5 GHz."

4. Also in the scan of the area, how long it takes to fully scan the area? What is the resolution?

Reply>> *We added this information at the end of the Methods section:*

"In the second experiment related to ZGV Lamb mode imaging, each frame of the animation has a spatial resolution of $\sim 2 \mu\text{m}$ and is recorded in somewhat over 2 min. The data acquisition for full animation requires ~ 12 h."

5. For the obtention of the dispersion curves, they perform a 2D FT analysis by making sweep along the x axis. Is this faster than the transient grating method? Once more, how long it takes to make the measurement?

Reply>> *In our case, for each pump-probe delay, each 1D spatial scan (containing 2001 recorded points) is performed in about 2 min, which is about 30% faster than in the case of the 2D images (as the number of points is reduced). We have added the following explanatory sentence in the Methods section, also comparing as requested the case of transient gratings:*

*"In the final experiment to determine dispersion curves, each 1D spatial scan (of the probe beam for a given pump-probe delay) is performed in ~ 2 min, leading to a total acquisition time ~ 8 h and a spatial resolution $\sim 2 \mu\text{m}$. This time is considerably longer than in a typical transient grating experiment [A. Maznev et al., Rev. Sci. Instrum. **74**, 667, 2003] owing to the need for the spatial scan in our case."*

6. I would acknowledge in general more details about the acquisition times in the experiments.

Reply>> *Exactly as the referee has requested, we believe we have properly explained all the relevant data acquisition times by the above corrections. Once again we thank the referee for pointing this out.*

7. For the spatial scan of the fields they have to move the probe, but in the set up they show in Fig.1a it is not clear to me how they can move the probe without moving the pump as well, which obviously is not enough to do this characterization, could the authors be more clear in this point?

Reply>> *The probe beam is directionally scanned by moving two mirrors, resulting in a slight lateral shift of the probe beam position on the dichroic mirror (indicated by the double-ended arrow in Fig. 1a). The angle of the beam entering the objective lens is thereby changed, resulting in a lateral displacement of the probe beam without affecting the pump.*

To make this clearer we have used two double-ended arrows in Fig. 1a, and have added the following sentence in the caption:

“The double-ended arrows represent the probe beam 2D scanning, realized by moving mirrors and a confocal lens pair (see Methods).”

And we have made it clearer in the Methods how this was done, including a reference:

*“A 2D spatial scanning system for the probe beam, composed of two moving mirrors and a confocal lens pair configuration [T. Tachizaki et al., Rev. Sci. Instrum. **77**, 043713, 2006], is mounted in the path of the probe beam together with a delay line to monitor the spatio-temporal evolution of the acoustic field at the sample surface up to ~10 GHz..”*

Reviewer #3 (Remarks to the Author):

Referee report

Manuscript ID: NCOMMS-18-36693

Tilte: “Imaging Gigahertz Zero-Group-Velocity Lamb Waves”

The authors report on the observation of zero-group-velocity Lamb modes in the hypersonic frequency range both in the time and space domain, by means of a state-of-the art ultrafast pump-probe optical microscopy. They exploit a smart pump and probe configuration allowing superior frequency resolution. The displacement field on a Ti-Si₃N₄ bilayer is spatially resolved with micrometric resolution. The zero-group-velocity Lamb mode life-time, Q-factor and dispersion relation are directly accessed.

The topic is timely and of high relevance. The results constitute a significant step forward in photoacoustic and bear great interests also for other communities. Indeed, I expect the methodology and the physics here shown will have a long-standing impact in fields such as mechanical nanometrology, non-destructive testing at the nanoscale and nanoscale-engineering of mechanical properties.

The manuscript is well written. The experimental evidence is exhaustive and its interpretation convincing. The data are clearly presented and thoroughly discussed. The authors furnish, through the SI and a tailored selection of the bibliography, all the elements needed to reproduce their work.

The manuscript fully complies with the criteria established for publication on Nature Communication. I therefore recommend publication of the present manuscript with minor revisions without need for further review.

Comments/suggestions.

I found the manuscript fully exhaustive both under a scientific and technical stand point. That said I have a couple of suggestions that might enlarge the interested audience.

In order to convey the significance of their message to other communities I suggest the authors to expand on their claim regarding their work potentials in the frame of mechanical metrology and non-destructive testing of nanomaterials, a claim which I strongly support.

The authors could:

(1) be more specific in addressing the “potential applications of real-time imaging of Gigahertz Zero-Group-Velocity Lamb Waves”, going beyond the few lines reported at the end of page 6 -beginning of page 7.

(2) Surface Acoustic Waves (in their broad sense) in the tens of GHz range and beyond have been recently proposed as a formidable tool for ultrathin film nanometrology. It would be nice if the authors could comment on the possibility of using their technique in conjunction with SAWs as a tool for non-destructive testing of nanometric films or multilayers.

Reply>> *Thank you very much for these two suggestions. Very similar points were raised by the Referee 1, and so we have significantly extended our conclusions to account for these suggestions, including extra references and a comparison with SAW techniques and SAW transient grating methods:*

*“Applications of real-time imaging of ZGV Lamb modes include the detection of defects in adhesion and deviations in interfacial stiffnesses, so this high frequency imaging technique should provide new avenues for evaluating and quantifying the mechanical integrity of nanostructures. In particular, our approach allows the probing of nanolayers. One example is the measurement of interfacial adhesion in two- or multi-layer nano-membrane systems. ZGV modes have already proven efficient in the determination of interfacial adhesion in layers of thickness in the mm range [Mezil et al., Appl. Phys. Lett. **105**, 021605, 2014; Mezil et al., J. Acoust. Soc. Am. **138**, 3202, 2015], so our work should encourage research into the evaluation of both normal and tangential interfacial adhesion in ultrathin multilayers of thickness in the nm range, complementing other GHz approaches to nanolayer adhesion metrology based on picosecond ultrasonics [Rossignol et al., Phys. Rev. B **70**, 094102, 2003; Antonelli et al., MRS Bulletin **31**, 607, 2006; Grossman et al., New J. Phys. **19**, 053019, 2017].*

*Moreover, since the use of finely-controlled arbitrary frequencies lends itself so well to the precise determination of resonances and their associated Q factors, it should prove possible to apply GHz ZGV imaging to the characterization of elastic constants in nanoscale layers in a similar way to that demonstrated in picosecond ultrasonics [Mante et al., Appl. Phys. Lett. **93**, 071909, 2008]. Our 2D imaging approach would also be particularly powerful for analysing complex directional behaviour when applied to nanoscale-thickness anisotropic layers—following the example of previous time-domain GHz surface-acoustic-wave imaging of crystals and phononic crystals [Sugawara et al., Phys. Rev. Lett. **88**, 185504, 2002; Otsuka et al., J. Appl. Phys. **117**, 245308, 2015]—including the possibility of directional excitation with laser line sources. The ability to image and thereby directionally distinguish and analyse the different in-plane ZGV acoustic modes would further facilitate*

*the extraction of elastic constants on the nanoscale by analogy with work on anisotropic ZGV modes in the MHz frequency range [Cès et al., J. Acoust. Soc. Am. **132**, 180, 2012].*

*Comparing our approach with that of surface-acoustic-wave transient gratings (TGs) in the same GHz frequency range (see, for example, [Bruchhausen et al., Phys. Rev. Lett. **106**, 077401, 2011] on the same sample would be fruitful, but especially our technique, exhibiting a much finer frequency control, should allow higher resolution of individual resonances and the detection of modes with larger Q factors. Finally, by extending the 2D time-domain imaging methods used for light propagation in optical waveguides and microstructures [Engelen et al., Nat. Phys. **3**, 401 EP-, 2007], it should be possible to image photonic ZGV modes in the time domain in a way analogous to the work presented here.”*

The x-y axis in Fig. 1 (a), which is referred to in the manuscript, is missing.

Reply>> *We thank the reviewer for noticing it. We added the axes on the new Fig. 1.*

REVIEWERS' COMMENTS:

Reviewer #1 (Remarks to the Author):

The authors have addressed my concerns and I find the revised manuscript suitable for publication in Nature Communications.

Reviewer #2 (Remarks to the Author):

The authors have properly answered my questions and clarified all them in the manuscript as well.

The paper is suitable for publication in NC in its current form.

Reviewer #3 (Remarks to the Author):

I recommend publication of the manuscript in its present form.